# Development of an Integrated Model for Open-Pit-Mine Discontinuous Haulage System Optimization

Miodrag Čelebić [1], Dragoljub Bajić [1,2,*], Sanja Bajić [2], Mirjana Banković [2], Duško Torbica [1], Aleksej Milošević [1] and Dejan Stevanović [2]

[1]  Faculty of Mining, University of Banja Luka, 79000 Prijedor, Republic of Srpska, Bosnia and Herzegovina; miodrag.celebic@rf.unibl.org (M.Č.); dusko.torbica@rf.unibl.org (D.T.); aleksej.milosevic@rf.unibl.org (A.M.)

[2]  Faculty of Mining and Geology, University of Belgrade, 11000 Belgrade, Serbia; sanja.bajic@rgf.bg.ac.rs (S.B.); mirjana.bankovic@rgf.bg.ac.rs (M.B.); dejan.stevanovic@rgf.bg.ac.rs (D.S.)

*  Correspondence: dragoljub.bajic@rgf.bg.ac.rs; Tel.: +38-164-668-4388

**Abstract:** The selection of the optimal equipment for discontinuous haulage systems is one of the most important decisions that need to be made when an open-pit mine is designed. There are a number of influencing factors, including natural (geological and environmental), technical, economic, and social. Some of them can be expressed numerically, in certain units of measure, while others are descriptive and can be stated by linguistic variables depending on the circumstances of the project. These factors are characterized by a high level of uncertainty, associated with both exploration and mining operations. The experience, knowledge, and expert judgment of engineers and specialists are of key importance for the management of mining processes, consistent with the issues stemming from the dynamic expansion of open-pit mines in space over time. This paper proposes an integrated model that translates all the criteria that affect the selection of the optimal solution into linguistic variables. By employing the multiple-criteria decision-making method and combining it with fuzzy logic, we developed an algorithm that addresses all the above-mentioned uncertainties inherent in various mining processes where the experience of experts forms the basis. The fuzzy analytic hierarchy process is used in order to deal with trending decision problems, such as mining equipment and management system selection. The entire algorithm was applied to a real case study—the Ugljevik East 1 open-pit mine.

**Keywords:** equipment selection; mine mechanization; expert judgment; linguistic variables; MCDM; FAHP

## 1. Introduction

Mining is one of the base industries and in many countries, a key sector of the economy [1,2]. The excavation and haulage systems of open-pit mines deliver millions of tons of useful material but also generate tens of billions of tons of waste rock per annum globally [3]. This amount of material is accompanied by many problems [4], ranging from geological to geopolitical. In recent times, the focus has been on environmental issues, including transition to clean energy aimed at implementing renewable energy technologies [5] and defining stringent carbon dioxide emission restrictions. This topic has been addressed in terms of both coals [6] and metallic ores [7,8].

Optimization of the entire system can determine whether a mining project is profitable or not. Each stage of the process needs to be optimized (open-pit limits [9], the mining method [10,11], the haulage system [12–14], the transportation equipment [15,16], the drilling and blasting pattern [17], etc.).

The excavation of material from open-pit mines is accomplished by continuous or discontinuous equipment, or a combination of the two. The type of equipment depends on a large number of factors [18]. A discontinuous system largely relies on an excavator and a number of dump trucks that haul the material [19]. Because of high capacities, excellent

flexibility, and relatively low operating and capital costs, excavators and dump trucks represent the most widely used load-and-haul method in open-pit mines [20]. Considerable attention has been paid to the factors that affect excavator performance [21], especially during the bucket cycle [22,23]. The most commonly used excavators are hydraulic and electrical rope shovels [24] because of their compactness and a broad range of capacities and possible bench heights. Another type of excavator that is often used is the dragline [25,26], especially for surface coal mining [27–29]. A special feature of this excavator is that it can operate independently (and transfer material) or work in tandem with dumpers that haul the material.

One of the leading challenges of mining system optimization (including optimization of the loading and haulage system) is the inability to consider criteria that cannot be expressed numerically. It might be easier to use natural language and express the criteria in linguistic variables, but their interaction needs to be determined. In jointly assessing criteria that can and cannot be expressed numerically, the most convenient approach is to apply multicriteria decision-making in combination with fuzzy logic.

Multicriteria decision-making methods take into account facts that are often ambiguous and imprecise and with uncertainty factors, to which the response has been (in mining, among other areas) to introduce fuzzy multicriteria decision-making (fuzzy MCDM).

An ever-present problem in mining practice is the optimization of the excavator–truck system, which requires proper excavator and truck selection for existing mining conditions [12,30]. In addition, optimization may focus on the selection of the truck type and number for a given type of excavator or on the performance of multiple excavators and several types of dump trucks [31]. The number of required bucket cycles is an important factor when choosing a truck. According to research, the optimal number is 3–6 bucket cycles [32]. The relationship between the capacities of the excavator and dump truck is defined by a match factor [31], which needs to be within an appropriate range.

Dump truck optimization can be divided into several stages, the first of which would be the choice of dump truck model for the given operating conditions [33,34]. The choice of truck size directly affects the road width, and to a lesser extent the road length, because of different minimum turning radii. Next is the selection of the number of trucks [35] and their distribution [30], as well as optimal routing from the point of loading to the point of unloading [36,37]. Haulage systems can also be compared in terms of energy consumption so that aspect can be included in equipment selection as well [38].

The primary goal of equipment selection optimization is to achieve the required capacity, which equipment of certain size and characteristics will allow. Additional objectives include safety at work, environmental protection, and profitability [20]. Total costs are one of the important parameters that affect equipment selection given that the operating cost of load-and-haul systems amount to 60% of the overall expenses of an open-pit mine [39]. If the same production capacity can be achieved with different types of haulage systems, then optimization boils down to optimizing costs.

The selection of load-and-haul equipment has been largely based on experience-proven methods, especially if the mine already operates certain models of excavators and trucks. However, as new variants of equipment with improved features are developed, selection methods need to be perfected [40].

Taking into account the above-mentioned challenges associated with mining processes, the objective of the paper is to describe and establish methodologies that can be used to select and design the optimal haulage system for the complex geological, technological, economic, and environmental conditions typical of open-pit mines. Generally speaking, the paper comprises three parts: problem description and introductory comments, description of methodology, and application to a real case study. The starting point of the research was the assumption that fuzzy MCDA can be used effectively to optimize the selection of mining equipment, specifically the type of dump truck for an existing excavator.

## 2. Methodology

Chang and Deng [41,42] describe a fuzzy approach to solving problems of qualitative multicriteria analyses as applied to bid selection and choice of employee candidates using different criteria. Inspired by their research, an algorithm was developed to address complex mining problems, such as the design of optimal machinery and excavator–truck systems. In general, the algorithm comprises three phases.

The first phase is the evaluation of conditions that will lead to adequate deployment of the type of equipment—dump trucks (options) for transportation management—and an analysis of effectiveness. The second phase includes the identification and detailed analysis of the factors that affect the selection of the optimal type of dump truck for a given excavator–truck system. The following factors were deemed to be universal: deposit and working area conditions, capital cost, operating cost, organizational complexity, and road infrastructure. The third phase evaluates the criteria and alternatives by fuzzy optimization and makes the final decision about the optimal type of dump truck for the excavator–truck system. In order to facilitate complex mathematical calculations associated with the determination of the optimal solution and the sensitivity analysis, an application specially developed for that purpose, FUZZY-GWCS® [43,44], was used in the third phase. Mathematical optimization calculations were based on the fuzzy-AHP extent analysis, namely the fuzzy analytic hierarchy process presented by Chang [41]. The mathematical optimization and decision-making procedures are described below according to the algorithm shown in Figure 1.

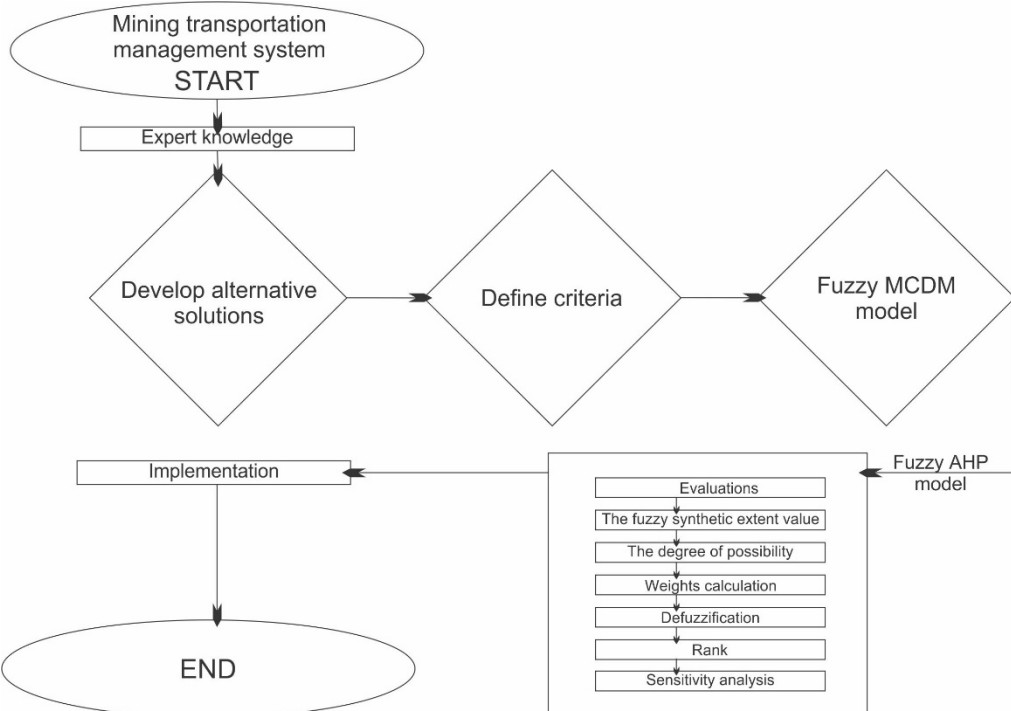

**Figure 1.** Mining transportation management system algorithm.

Experts face diverse problems in surface mining investigations, both geological and those associated with the management of different operations and processes. Successful development and design of alternative solutions require a large amount of knowledge in multiple areas of expertise, such as geology, hydrogeology/dewatering, rock mechanics, construction of drains, mining methods, transportation systems, machinery, and management processes in mining. As such, experts conduct various types of analyses of the factors inherent in open-pit mining in order to fully define their characteristics. A quality analysis of all these factors directly influences the efficiency of designing a mining transportation management system.

After expert judgments are analyzed, alternative solutions are generated and the criteria that govern the selection of the optimal solution are identified.

The process continues with the creation of a fuzzy MCDM model (i.e., a fuzzy-AHP model). This is a long process with several steps and mathematical procedures, which are repeated as required by the set hierarchy. Matrices are created and the criteria evaluated against each other and also relative to the alternative solutions. A scale of relative importance [45], or the so-called fuzzified scale [41,42,46], is used for this purpose, where there is a connection between numerical values of triangular fuzzy numbers and linguistic variables. Each element of the matrix is evaluated to formulate a question for the expert who is studying the problem, namely, "Is one criterion more important than another in a pairwise comparison and, if so, to what extent?"

The next step is determining the fuzzy synthetic degree value as follows. Let $X = \{x_1, x_2, \ldots, x_n\}$ be the analyzed set and $G = \{g_1, g_2, \ldots g_n\}$ the target set. An extent analysis is conducted for all the elements of set $X$ and each element of set $G$ [47]. This results in $m$ extent analysis values for each element of set $X$ as follows:

$$M_{g_i}{}^1, M_{g_i}{}^2, \ldots, M_{g_i}{}^m, \ i = 1, 2, \ldots, n$$

where all of $M_{g_i}{}^j$, $j = 1, 2, \ldots, m$ are triangular fuzzy numbers.

Let $M_{g_i}{}^1, M_{g_i}{}^2, \ldots, M_{g_i}{}^m$ signify the extent analyses of the elements of those sets for $m$. In this case, the fuzzy synthetic degree values (for $i$ elements) are calculated as follows (Equation (1)):

$$S_i = \sum_{j=1}^{m} M_{g_i}{}^j \otimes \left[ \sum_{i=1}^{n} \sum_{j=1}^{m} M_{g_i}{}^j \right]^{-1}, \tag{1}$$

Otherwise, if triangular fuzzy numbers of the form $M = (l, s, d)$ are considered, then the following is applied (Equations (2)–(5)):

$$(M_1 = (l_1, s_1, d_1), \ M_2 = (l_2, s_2, d_2) \ldots) : \tag{2}$$

$$\sum_{j=1}^{m} M_{g_i}^j = \left( \sum_{j=1}^{m} l_j, \sum_{j=1}^{m} s_j, \sum_{j=1}^{m} d_j \right) \tag{3}$$

$$\sum_{i=1}^{n} \sum_{j=1}^{m} M_{g_i}{}^j = \left( \sum_{i=1}^{n} l_i, \sum_{i=1}^{n} s_i, \sum_{i=1}^{n} d_i \right), \tag{4}$$

where $M_{g_i}{}^j (j = 1, 2, \ldots n)$

$$\left[ \sum_{i=1}^{n} \sum_{j=1}^{m} M_{g_i}^j \right]^{-1} = \left( \frac{1}{\sum_{i=1}^{n} d_j}, \frac{1}{\sum_{i=1}^{n} s_j}, \frac{1}{\sum_{i=1}^{n} l_j} \right) \tag{5}$$

Ultimately, the fuzzy synthetic degree value is expressed as follows (Equation (6)):

$$S_i = \left( \sum_{j=1}^{m} l_j, \sum_{j=1}^{m} s_j, \sum_{j=1}^{m} d_j \right) \otimes \left( \frac{1}{\sum_{i=1}^{n} d_j}, \frac{1}{\sum_{i=1}^{n} s_j}, \frac{1}{\sum_{i=1}^{n} l_j} \right) \tag{6}$$

The next step is to determine the degree of possibility. The first task of the fuzzy-AHP process is to decide on the relative importance of each pair of factors in the same hierarchy.

A fuzzy matrix $A = (a_{ij})_{n \times m}$ is created using triangular fuzzy numbers and making a pairwise comparison (of elements), where $a_{ij} = (l_{ij}, s_{ij}, d_{ij})$. This satisfies the following condition (Equation (7)):

$$l_{ij} = \frac{1}{l_{ji}}, s_{ij} = \frac{1}{s_{ji}}, d_{ij} = \frac{1}{d_{ji}} \tag{7}$$

The final step of the FAHP analysis is determining the weight priority vector of each criterion. This requires consideration of the fuzzy number comparison principles, or a "min" and "max" strategy operation. Based on the above, the degree of possibility of two

fuzzy numbers is determined by applying the principle of fuzzy number comparison, as described below.

If two triangular fuzzy numbers, $M_1 \geq M_2$, are compared, then the degree of possibility can be described as follows (Equation (8)):

$$V(M_1 \geq M_2) = \sup_{x \geq y}\left[min\left(\mu_{M_1}(x), \mu_{M_2}(y)\right)\right] \tag{8}$$

where if there are $(x, y)$ pairs, such that $x \geq y$ and $\mu_{M_1}(x) = \mu_{M_2}(y) = 1$, then $V(M_1 \geq M_2) = 1$. Given that $M_1$ and $M_2$ are convex triangular fuzzy numbers, the following can be applied (Equations (9) and (10)):

$$V(M_1 \geq M_2) = 1 \ if \ s_1 \geq s_2 \tag{9}$$

$$V(M_2 \geq M_1) = 1 = hgt(M_1 \cap M_2) = \mu_{M_1}(c) \tag{10}$$

where $c$ is the ordinate of the highest intersection point $C$ between the membership functions $\mu_{M_1}$ and $\mu_{M_2}$ (Figure 2).

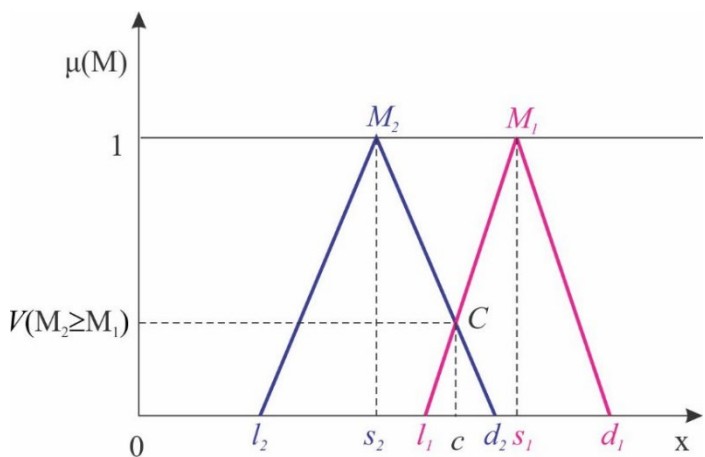

**Figure 2.** Triangular fuzzy numbers.

Finally, the degree of possibility for the triangular fuzzy numbers $M_1 = (l_1, s_1, d_1)$ and $M_2 = (l_2, s_2, d_2)$ can equally be expressed as follows (Equation (11), point C):

$$V(M_2 \geq M_1) = hgt(M_1 \cap M_2) = d_{M_2}(c) = \begin{cases} 1, & if \ s_2 \geq s_1 \\ 0, & if \ l_1 \geq d_2 \\ \frac{l_1 - d_2}{(s_2 - d_2) - (s_1 - l_1)}, & otherwise \end{cases} \tag{11}$$

Both values, $V(M_1 \geq M_2)$ and $V(M_2 \geq M_1)$, are needed to compare triangular fuzzy numbers $M_1$ and $M_2$.

The degree of possibility for a convex triangular fuzzy number to be greater than $k$ of convex fuzzy number $M_i$, where $i = 1, 2, \ldots, k$, can be defined as follows (Equation (12)):

$$V(M \geq M_1, M_2, \ldots M_k) = V[(M \geq M_1) \wedge (M \geq M_2) \wedge \ldots \wedge (M \geq M_k)] = minV(M \geq M_i) \tag{12}$$

The above leads to the following (Equation (13)):

$$c'(A_i) = minV(S_i \geq S_k), \ k = 1, 2, \ldots, n; \ k \neq i \tag{13}$$

The next step is to define the weight priority vector, as follows (Equation (14)):

$$W' = \left(c'(A_1), c'(A_2), \ldots, c'(A_n)^T\right) \text{ where } A_i(i = 1, 2, \ldots n) \tag{14}$$

Following normalization, the normalized weight priority vector is as follows (Equation (15)):

$$W = \left( c(A_1), c(A_2), \ldots, c(A_n)^T \right) \tag{15}$$

where $W$ is a defuzzified, conventional non-fuzzy number.

As mentioned at the beginning, these mathematical operations are undertaken to compare the criteria to each other, as well as compare the alternatives separately for each criterion. This results in matrices and weight priority vectors.

The final weights of the alternatives are calculated at the end of the mathematical optimization operations. They are derived by additive aggregation, namely by multiplying the weight priority vectors from the criteria matrix by the weight priority vectors calculated in the evaluation of the alternatives relative to all the criteria. The alternative with the highest value of the weight priority vector is the best choice.

On the other hand, a sensitivity analysis can be undertaken by introducing the optimization index λ and calculating the total integral values—$I$—which results in the weights of the alternatives that reflect the risk assessment of the expert as follows (Equation (16)) [48,49]:

$$I = \frac{(d\lambda + s + (1 - \lambda)l)}{2}, \quad \lambda \in [0, 1] \tag{16}$$

In the above equation, $l$, $s$, and $d$ stand for triangular fuzzy number parameters (Figure 2). For the optimization index, a greater value is indicative of a higher degree of optimism. The scientists mentioned in the paper generally take the following values: 0 for pessimistic, 0–5 for moderate, and 1 for optimistic.

Experts ultimately sublimate the entire algorithm-based analysis and produce a multiyear mining transportation management system plan. If implemented successfully, ore mining and other mine management processes are systematized and simplified.

## 3. Case Study

The study area for the proposed algorithm was the open-pit mine Ugljevik East 1, located in the northeastern part of Bosnia and Herzegovina (Figure 3). Coal is mined at Ugljevik East 1 for the thermal power plant (TPP) Ugljevik.

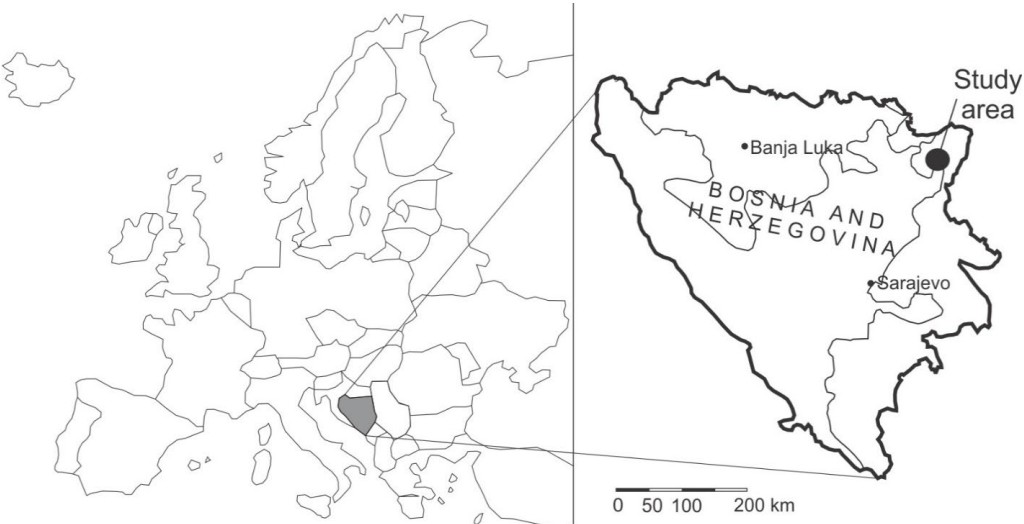

**Figure 3.** Geographical map of the study area.

In terms of genesis and sedimentation, the Ugljevik East 1 coal deposit falls within the central part of the Ugljevik coal-bearing zone. According to the lithology (Figure 4), paleontology, and superposition, the geologic framework of the area comprises Paleocene–Eocene deposits, a complex of freshwater coal-bearing sediments, and Tortonian marine

deposits. Ugljevik East 1 is a continuous extension of the Ugljevik coal-bearing formation toward the east. In its southern part, the spread of the productive part of the formation was discontinued by tectonic activity, which resulted in uplifting and exposed this part of the formation to erosion. Coal is found in three to five seams, whose structure is highly complex. The seams trend north–northeast at an angle of about 20° (or more in the northern district). The coal at Ugljevik East 1 is of the brown (lignite) type. The thickness of the main coal seam is mostly 15 to 20 m but can be up to 38.8 m in some places.

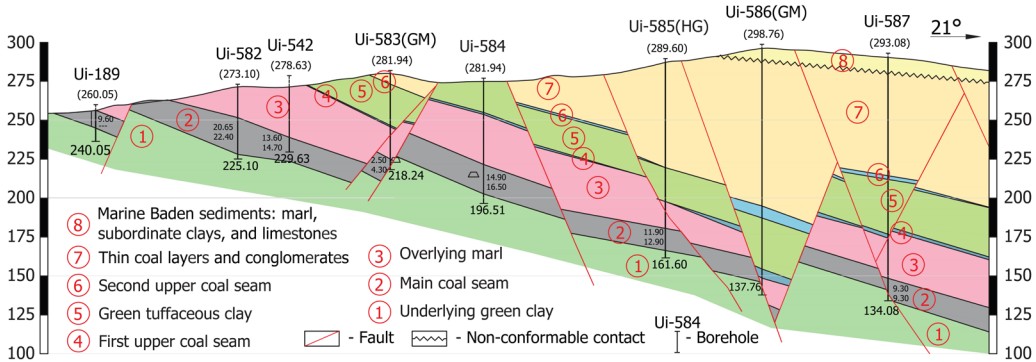

**Figure 4.** General geological cross-section of the Ugljevik East 1 deposit.

The production capacity ($1.8 \times 10^6$ t/year) and identified coal reserves can support mining over the next 20 years. Apart from the coal, some $23 \times 10^6$ tons of waste is excavated at Ugljevik East 1. The production system at the mine comprises the following:

1. Excavation, haulage, crushing, and deposition of the coal at the TPP;
2. Excavation, haulage, and disposal of the waste rock.

The entire coal-mining process also includes transportation by a system of belt conveyors to the TPP, as shown in Figure 5. The coal is mined by three excavators, two Komatsu PC 1250SP and one Liebherr R974B. The dump trucks are Belaz (Žodzina, Belarus) of 90 t payload capacity.

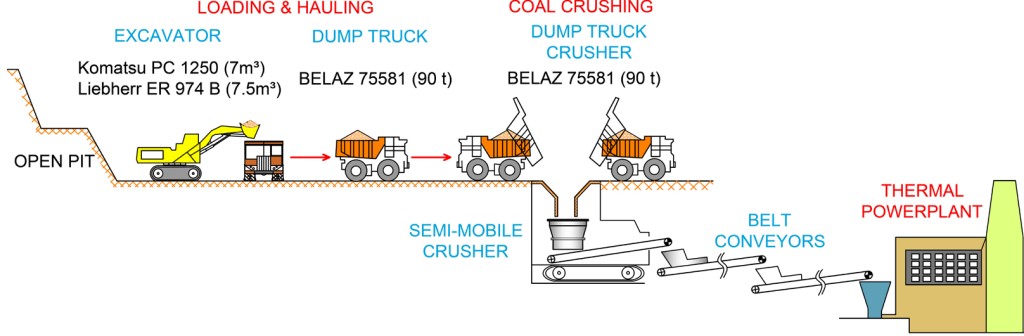

**Figure 5.** Current coal-mining technology at Ugljevik East 1.

As shown in Figure 6, the waste is currently transported by three types of dump trucks. This situation is not optimal because of the different characteristics of the trucks (payload capacities, speeds, loading times, size of working areas, etc.), which affect the productivity of the entire system. Dump truck standardization tends to play a key role in servicing and repair-cost reduction [50]. With all these factors being considered, the type of waste dump trucks was optimized in this research. Given the positive experience gained so far, as well as the fact that there are currently five Komatsu PC3000 excavators at Ugljevik East 1, the excavator type was not examined. The study focused only on the selection of the type of dump truck.

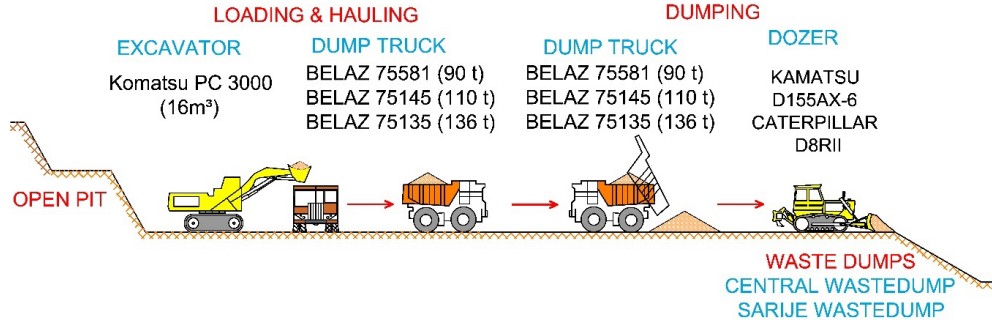

**Figure 6.** Current waste-mining technology at Ugljevik East 1.

## 4. Results and Discussion

Given past experience and the existing infrastructure with regard to dump truck maintenance, repair, and management, all the considered trucks are made by Belaz. Four alternative solutions, or four types of dump trucks, were examined to optimize equipment selection:

- Alternative 1 (A1): Belaz 75581 (payload capacity 90 t);
- Alternative 2 (A2): Belaz 75145 (110 t);
- Alternative 3 (A3): Belaz 75135 (136 t);
- Alternative 4 (A4): Belaz 7517 (160 t).

The range of payload capacity (90 t to 160 t) was evaluated with respect to the size of the mine, the properties of the coal deposit, and the required annual capacity of waste-rock loading and haulage ($22.3 \times 10^6$ t/year). The main characteristics of the considered trucks are shown in Table 1.

**Table 1.** Main characteristics of the considered dump trucks.

| Alternative | Payload Capacity (t) | Gross Truck Weight (t) | Body Volume Heaped 2:1 (m³) | Engine Power (kW) | Maximal Speed (km/h) | Truck Width (m) | Turning Radius (m) |
|---|---|---|---|---|---|---|---|
| A1 | 90 | 164 | 53.3 | 895 | 60 | 5.36 | 11 |
| A2 | 110 | 210 | 67 | 1194 | 64 | 6.4 | 13 |
| A3 | 136 | 243 | 80 | 1194 | 50 | 6.4 | 13 |
| A4 | 160 | 294 | 96.5 | 1492 | 65.6 | 6.9 | 14 |

To arrive at the optimal solution, dump truck performance was analyzed against several criteria, including production performance, deposit and working-area conditions, capital cost, operating cost, organizational complexity, and road infrastructure. It should be noted that these criteria tend to be universal and applicable to many open-pit mines, albeit with some adjustment of the relationships between the individual criteria depending on the mining conditions. This was the approach followed in the assessment of the criteria and their effect in accordance with the specific conditions at Ugljevik East 1. The criteria that influenced the optimization of the dump truck type are briefly described below.

Production performance relates to the ability of each of the examined dump truck types to achieve the given annual waste haulage capacity. Talpac–3D software 3.8 [51] was used to simulate the operation of the PC3000 excavator with each of the four dump truck types, providing data on production per operating hour and the number of needed trucks. The results of the simulation showed that the production objectives can be achieved with each of the four dump truck types, but the number of needed trucks varied depending on truck productivity (Table 2).

**Table 2.** Productivity and number of needed dump trucks.

| Alternative | Payload Capacity (t) | Engine Power (kW) | Production per Operating Hour (t/h) | Number of Needed Trucks |
|---|---|---|---|---|
| A1 | 90 | 895 | 137 | 29 |
| A2 | 110 | 1194 | 151 | 27 |
| A3 | 136 | 1194 | 172 | 25 |
| A4 | 160 | 1492 | 244 | 19 |

Road infrastructure is a factor that influences, to a large extent, the selection of a suitable dump truck type and size. The road infrastructure and dump truck type need to be compatible in order to improve productivity and minimize production risks [52]. Dump trucks whose payload capacity is higher and whose size is consequently larger will require wider roads and larger turning radii, as well as a higher load-bearing capacity of the road. Such roads are more expensive due to the higher quality and larger quantity of materials, higher maintenance spending, and the greater number of auxiliary equipment units required. This analysis is based on the cost of constructing one meter of road, which is governed by payload capacity (affecting the load bearing capacity of the road) and the width of the truck (i.e., the width of the road). Table 3 shows the results of the road infrastructure analysis relative to dump truck type.

**Table 3.** Cost of constructing one meter of road length depending on dump truck type.

| Alternative | Payload Capacity (t) | Truck Width (m) | Road Width (m) | Road Cost per 1 m of Length (€/m′) |
|---|---|---|---|---|
| A1 | 90 | 5.36 | 16.1 | 242 |
| A2 | 110 | 6.4 | 19.2 | 288 |
| A3 | 136 | 6.4 | 19.2 | 288 |
| A4 | 160 | 6.9 | 20.7 | 311 |

Organizational complexity grows significantly as the number of units engaged in waste haulage increases. A large number of haulage units have an adverse effect on work organization given that equipment productivity may be impaired by bottlenecks, delays, and additional losses during maneuvering at loading and unloading points. A complex organization of operations requires considerable logistical support and encumbers both maintenance and data and cost management [53]. Given that the organizational complexity increases with the number of dump trucks, this criterion favors larger dump trucks. However, a greater number of smaller dump trucks increase flexibility, which is especially important where scheduling adjustments are nee, or where the application of selective mining techniques is necessary (along structurally complex zones with a limited working area).

Deposit and working-area conditions, such as the type, depth, angle, and engineering geology characteristics of the deposit, have a significant effect on the selection of the mining equipment types and sizes. Even though the focus of this research was on the selection of the optimal dump truck for waste, the production conditions are largely determined by the structure of the coal seams. In this regard, it should be noted that the geological structure of the Ugljevik East 1 coal deposit is largely determined by the presence of numerous faults (Figure 4) and the consequent high complexity (steep seam, extensive layering, loss of geological continuity). This kind of structure impedes mining from slope-stability and bearing-capacity perspectives and requires selective excavation. Such excavation of coal and overburden along zones of often small size due to the presence of faults favors small-size equipment, whereas the need to excavate large amounts of waste (high overburden coefficient) gives preference to high-capacity equipment [54].

Operating cost has a major effect on the profitability, sustainability, and efficiency of mining operations. It includes expenses associated with labor and equipment operation and maintenance. They are incurred throughout the life cycle of the equipment and largely depend on the equipment size, number of units in operation, extent of use, and quality of maintenance. In the present case study, the operating cost estimate was based on past experience from Ugljevik East 1 and equipment manufacturer's assessments [55].

Table 4 shows the labor cost by dump truck type and number. Based on data obtained from the mine, the estimated gross labor cost of a single truck driver is EUR 2000 per month. The labor costs in Table 4 favor larger dump trucks because they are more productive (i.e., fewer dump trucks and fewer drivers needed).

**Table 4.** Labor costs by type of dump truck.

| Alternative | Payload Capacity (t) | Number of Needed Trucks | Number of Needed Drivers | Labor Cost per Year (EUR/year) |
|---|---|---|---|---|
| A1 | 90 | 29 | 145 | 3,480,000 |
| A2 | 110 | 27 | 135 | 3,240,000 |
| A3 | 136 | 25 | 125 | 3,000,000 |
| A4 | 160 | 19 | 95 | 2,280,000 |

The cost of materials is defined based on parameters such as engine power, fuel, lubricants, and tires. They were calculated per ton of waste and then multiplied by the total planned annual capacity for waste ($23 \times 10^6$ t) to obtain the total annual cost of materials. Table 5 shows the total cost of materials by type of dump truck.

**Table 5.** Cost of materials by type of dump truck.

| Alternative | Payload Capacity (t) | Engine Power (kW) | Fuel (EUR/t) | Lube (EUR/t) | Tires (EUR/t) | Total (EUR/t) | Total per Year (EUR/Year) |
|---|---|---|---|---|---|---|---|
| A1 | 90 | 895 | 0.520 | 0.026 | 0.016 | 0.562 | 12,917,000 |
| A2 | 110 | 1194 | 0.661 | 0.033 | 0.020 | 0.714 | 16,420,000 |
| A3 | 136 | 1194 | 0.580 | 0.029 | 0.017 | 0.627 | 14,415,000 |
| A4 | 160 | 1492 | 0.511 | 0.026 | 0.015 | 0.552 | 12,698,000 |

Capital cost refers to the procurement of equipment and is a very important factor of the planning process. The objective is to achieve a balance between investment in and productivity of mining equipment. Large expenditures, such as for the purchasing of large dump trucks with a high payload capacity, often require unfavorable bank loans and have an adverse effect on the economics of a project. On the other hand, the purchase price of smaller dump trucks will be lower but so will the capacity. In addition, a larger number of units will be required. As a result, smaller dump trucks, with a lower payload capacity, can often generate a higher capital cost. The optimization of financial performance depends to a large extent on efficient capital cost management coupled with the achievement of an appropriate level of equipment productivity and reliability. Capital cost estimates are generally based on the equipment buyer's requirements and market research [56]. The following parameters were analyzed in this regard: dump truck payload capacity, unit cost, and the required number of dump trucks. Table 6 shows the total capital cost by the type of dump truck.

As mentioned above, the range of payload capacity (from 90 t to 160 t) was consistent with the size of the open-pit mine, the properties of the coal deposit, and annual requirements relating to waste loading and haulage. The decision to consider dump trucks made by Belaz was justifiable according to past experience and the existing infrastructure for vehicle maintenance, repair, and management.

**Table 6.** Capital cost of dump truck procurement.

| Alternative | Payload Capacity (t) | Purchase Price per Unit (EUR) | Number of Needed Trucks | Total Truck Capital Cost (EUR) |
|---|---|---|---|---|
| A1 | 90 | 1,270,000 | 29 | 36,830,000 |
| A2 | 110 | 1,400,000 | 27 | 37,800,000 |
| A3 | 136 | 1,600,000 | 25 | 40,000,000 |
| A4 | 160 | 1,770,000 | 19 | 33,630,000 |

The optimal solution in the present case was based on analyses of the performance of the excavator–truck system against the following criteria:

Production performance;
Road infrastructure (K1);
Organizational complexity (K2);
Deposit and working-area conditions (K3);
Operating cost (K4);
Capital cost (K5).

The production performance criterion was analyzed in order to determine whether each of the dump truck types could be used to haul waste. This analysis, completed by means of Talpac software [51], showed that in technical terms, all the considered types would be capable of handling the given volume of waste. After production performance was assessed for each dump truck type, optimization proceeded with analyses of the other five criteria.

The conditions prevailing at Ugljevik East 1 needed to be defined in order to assess the other criteria. These criteria would not affect each other equally at another open-pit mine. In effect, the conditions and their impact are individual features of each mine.

The fuzzy optimization methodology described above was followed using the identified criteria and alternatives. The calculations were made applying the specially developed software FUZZY-GWCS [43,44]. The inputs were numerical values of linguistic variables, represented by triangular fuzzy numbers. Table 7 shows the values of the criteria matrices and the calculated values of their weight priority vectors.

**Table 7.** Analysis of criteria.

| Criterion | K1 | | | K2 | | | K3 | | | K4 | | | K5 | | | Values of Weight Coefficients | | |
|---|---|---|---|---|---|---|---|---|---|---|---|---|---|---|---|---|---|---|
| K1 | 1 | 1 | 1 | 1 | 2 | 3 | 2 | 3 | 4 | 4 | 5 | 6 | 5 | 6 | 7 | 0.215 | 0.348 | 0.550 |
| K2 | 0.33 | 0.50 | 1 | 1 | 1 | 1 | 1 | 2 | 3 | 4 | 5 | 6 | 5 | 6 | 7 | 0.187 | 0.297 | 0.472 |
| K3 | 0.25 | 0.33 | 0.50 | 0.33 | 0.50 | 1 | 1 | 1 | 1 | 3 | 4 | 5 | 4 | 5 | 6 | 0.142 | 0.222 | 0.354 |
| K4 | 0.17 | 0.20 | 0.25 | 0.17 | 0.20 | 0.25 | 0.20 | 0.25 | 0.33 | 1 | 1 | 1 | 2 | 3 | 4 | 0.059 | 0.095 | 0.153 |
| K5 | 0.14 | 0.17 | 0.20 | 0.14 | 0.17 | 0.20 | 0.17 | 0.20 | 0.25 | 0.25 | 0.33 | 0.50 | 1 | 1 | 1 | 0.028 | 0.038 | 0.056 |

Table 8 shows the evaluation of each alternative relative to each criterion. It also includes the values of weight priorities.

Following evaluation, the final values of all the alternatives were calculated in the form of triangular fuzzy numbers as were the final "weights" of the alternatives as non-fuzzy numbers and the optimization indices shown in Table 9. Based on the interpreted results, the largest "weight" was the "best" score. Alternative 4 (Belaz 7517 truck, payload capacity 160 t) was proposed as the best choice—the optimal haulage system. The runner up was Alternative 2, and the least favorable solution was Alternative 1.

Figure 7 shows the total integral value for a moderate, pessimistic, and optimistic expert's risk assessment and the weights of the alternatives relative to the optimization index. For an optimistic assessment ($\alpha = 1$) of the decision-maker, the weights of the alternatives vary over a very narrow range as compared to the pessimistic ($\alpha = 0$) and moderate ($\alpha = 0.5$) assessments. Based on the sensitivity analyses of all the alternatives, on

average, the differences in weight vary up to 1.03% for an optimization index of 0.5 and up to 5.64% for an optimization index of 0.

**Table 8.** Analysis of alternatives relative to criteria.

| Criterion | A1 | | | A2 | | | A3 | | | A4 | | | Values of Weight Coefficients | | |
|---|---|---|---|---|---|---|---|---|---|---|---|---|---|---|---|
| K1 | | | | | | | | | | | | | | | |
| A1 | 1 | 1 | 1 | 0.25 | 0.33 | 0.50 | 0.25 | 0.33 | 0.50 | 0.17 | 0.20 | 0.25 | 0.057 | 0.081 | 0.130 |
| A2 | 2 | 3 | 4 | 1 | 1 | 1 | 1 | 1 | 1 | 0.33 | 0.50 | 1 | 0.148 | 0.241 | 0.404 |
| A3 | 2 | 3 | 4 | 1 | 1 | 1 | 1 | 1 | 1 | 0.33 | 0.50 | 1 | 0.148 | 0.241 | 0.404 |
| A4 | 4 | 5 | 6 | 1 | 2 | 3 | 1 | 2 | 3 | 1 | 1 | 1 | 0.239 | 0.437 | 0.750 |
| K2 | | | | | | | | | | | | | | | |
| A1 | 1 | 1 | 1 | 2 | 3 | 4 | 4 | 5 | 6 | 7 | 8 | 9 | 0.342 | 0.494 | 0.711 |
| A2 | 0.25 | 0.33 | 0.50 | 1 | 1 | 1 | 2 | 3 | 4 | 5 | 6 | 7 | 0.202 | 0.300 | 0.445 |
| A3 | 0.17 | 0.20 | 0.25 | 0.25 | 0.33 | 0.50 | 1 | 1 | 1 | 3 | 4 | 5 | 0.108 | 0.161 | 0.240 |
| A4 | 0.11 | 0.13 | 0.14 | 0.14 | 0.17 | 0.20 | 0.20 | 0.25 | 0.33 | 1 | 1 | 1 | 0.035 | 0.045 | 0.059 |
| K3 | | | | | | | | | | | | | | | |
| A1 | 1 | 1 | 1 | 0.20 | 0.25 | 0.33 | 0.17 | 0.20 | 0.25 | 0.13 | 0.14 | 0.17 | 0.037 | 0.048 | 0.064 |
| A2 | 3 | 4 | 5 | 1 | 1 | 1 | 0.25 | 0.33 | 0.50 | 0.17 | 0.20 | 0.25 | 0.111 | 0.166 | 0.249 |
| A3 | 4 | 5 | 6 | 2 | 3 | 4 | 1 | 1 | 1 | 0.20 | 0.25 | 0.33 | 0.181 | 0.277 | 0.418 |
| A4 | 6 | 7 | 8 | 4 | 5 | 6 | 3 | 4 | 5 | 1 | 1 | 1 | 0.352 | 0.509 | 0.738 |
| K4 | | | | | | | | | | | | | | | |
| A1 | 1 | 1 | 1 | 0.14 | 0.17 | 0.20 | 0.20 | 0.25 | 0.33 | 1 | 2 | 3 | 0.061 | 0.108 | 0.180 |
| A2 | 5 | 6 | 7 | 1 | 1 | 1 | 3 | 4 | 5 | 5 | 6 | 7 | 0.365 | 0.538 | 0.793 |
| A3 | 3 | 4 | 5 | 0.20 | 0.25 | 0.33 | 1 | 1 | 1 | 3 | 4 | 5 | 0.188 | 0.293 | 0.449 |
| A4 | 0.33 | 0.50 | 1 | 0.14 | 0.17 | 0.20 | 0.20 | 0.25 | 0.33 | 1 | 1 | 1 | 0.044 | 0.061 | 0.100 |
| K5 | | | | | | | | | | | | | | | |
| A1 | 1 | 1 | 1 | 0.33 | 0.50 | 1 | 0.20 | 0.25 | 0.33 | 3 | 4 | 5 | 0.127 | 0.200 | 0.329 |
| A2 | 1 | 2 | 3 | 1 | 1 | 1 | 0.20 | 0.25 | 0.33 | 3 | 4 | 5 | 0.146 | 0.253 | 0.418 |
| A3 | 3 | 4 | 5 | 3 | 4 | 5 | 1 | 1 | 1 | 4 | 5 | 6 | 0.309 | 0.488 | 0.762 |
| A4 | 0.20 | 0.25 | 0.33 | 0.20 | 0.25 | 0.33 | 0.17 | 0.20 | 0.25 | 1 | 1 | 1 | 0.044 | 0.059 | 0.086 |

**Table 9.** Ranking and selection of the optimal approach.

| | Fuzzy Number | | | Value of Weight | Final | Sensitivity Analysis | | |
|---|---|---|---|---|---|---|---|---|
| | L | S | D | Priority Vector | Ranking | $\alpha = 0.0$ | $\alpha = 0.5$ | $\alpha = 1.0$ |
| A1 | 0.014 | 0.068 | 0.340 | 0.194 | 4 | 0.206 | 0.197 | 0.195 |
| A2 | 0.017 | 0.090 | 0.476 | 0.271 | 2 | 0.271 | 0.271 | 0.271 |
| A3 | 0.015 | 0.080 | 0.426 | 0.242 | 3 | 0.239 | 0.241 | 0.242 |
| A4 | 0.017 | 0.096 | 0.517 | 0.294 | 1 | 0.284 | 0.291 | 0.293 |

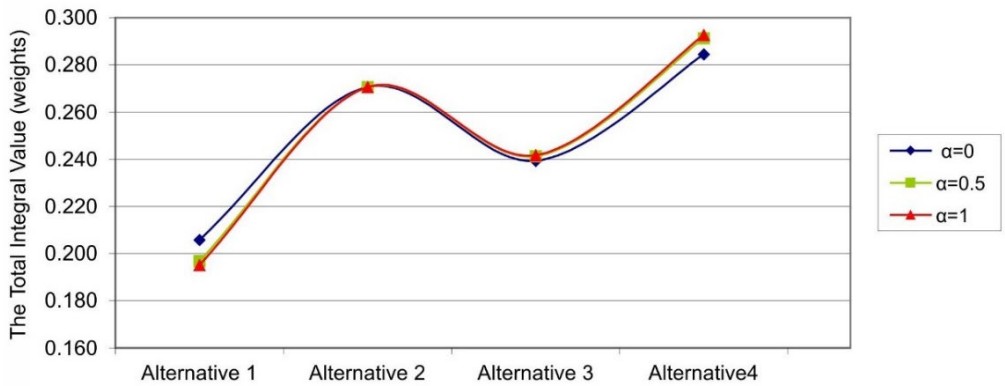

**Figure 7.** Total integral values of the moderate, pessimistic, and optimistic expert's risk assessments.

## 5. Conclusions

The selection of optimal equipment for a discontinuous haulage system is a complex task of mining engineering. Decision-making requires reliable knowledge about all the

required parameters of open-pit mining operations. This paper discusses all the criteria for selecting the optimal dump truck for a predefined type of excavator. The criteria used for the optimization model were based on the characteristics of the ore deposit, the work conditions, the road infrastructure, the operational complexity, and the operating and capital costs.

The FAHP method was used to create an integrated model capable of optimizing the selection of optimal equipment for a discontinuous haulage system. Modeling included defining of criteria of relevance to the loading and haulage operations, as well as options. This created the conditions for further calculations that provided alternatives and ultimately the optimal solution. The model combines multiple factors (criteria) that can be expressed numerically and factors that are descriptive, which are expressed by linguistic variables. FAHP is characteristic in that it solves problems in stages until the target is reached and is based on expert judgment and assessment of priorities.

After the choice of dump trucks for an excavator-truck system at an open-pit coal mine was optimized, Alternative A4 (Belaz 7517 truck, 160 t) was found to be optimal. The result indicated that future procurement should be focused on larger trucks for overburden haulage than for coal transportation.

The model was very efficient for dump truck selection for optimizing the excavator–truck system. It can be applied to many open-pit mines where the type of truck needs to be selected or where there is already an excavator–truck system in place that requires integrated insight into as many factors of influence as possible, which cannot all be expressed numerically for comparison purposes. In the presented case study, the model could be used to determine the optimal alternative for the renewal of an overburdened dump truck fleet or the unification of truck size.

The model has many advantages, but its application largely depends on the parameters associated with an open-pit mine. As such, continuous production monitoring and data collection are required and recommended. The presented study can be useful for defining the evaluation criteria for an existing open-pit mine or for selecting and optimizing excavator–truck systems in new mines with similar operating conditions.

The proposed model is universal in that it can be applied to all open-pit mines where an excavator–truck system is used, with some adjustments to the evaluation criteria being needed to reflect the specific case.

Contemporary conditions in the mining industry are characterized by a continuous decline in the quality of deposits (reduced mineral content, unfavorable structural characteristics of deposits, greater mining depths, etc.), progressively restrictive environmental and other administrative norms, and a turbulent sociopolitical environment. These changing business conditions necessitate ongoing research and the development of enhanced optimization methods. In this regard, future research should focus on the proper selection and assessment of influential criteria, which, with the help of advanced mathematical tools (optimization algorithms), are capable of determining the choice of an optimal solution from a set of considered alternatives.

Despite the similarities among many mining projects, the set of influential criteria is a unique characteristic of each location. This means that not only will different sets of criteria be relevant, but their impact will also vary depending on the nature of the problem, i.e., the characteristics of specific mining projects. For this reason, parameters related to mining production, which affect evaluation criteria, must be continuously collected, statistically processed, and systematically analyzed. Based on the gathered information, a database needs to be formed to assist in selecting relevant parameters and in accurately assessing their values. Optimization of a technological process should be re-executed as needed based on changes in the values of influential parameters. The presented mathematical model must be continuously improved to be robust enough to encompass all influential parameters and thereby ensure the generation of optimal solutions.

The presented interdisciplinary approach that connects surface mining (specifically mining mechanization) with fuzzy optimization contributes to the sustainable and improved handling of optimal equipment selection in mine management.

**Author Contributions:** Conceptualization, M.Č., D.S. and D.B.; methodology, D.B. and S.B.; software, M.Č.; validation, M.Č. and D.B.; formal analysis, D.S. and A.M.; investigation, M.Č., D.T. and A.M.; resources, M.Č., D.T. and A.M.; data curation, D.T. and M.B.; writing—original draft preparation, M.Č.; writing—review and editing, D.B., D.S. and S.B.; visualization, M.Č., D.S. and M.B.; supervision, D.B. All authors have read and agreed to the published version of the manuscript.

**Funding:** This research received no external funding.

**Institutional Review Board Statement:** Not applicable.

**Informed Consent Statement:** Not applicable.

**Data Availability Statement:** Data are contained within the article.

**Acknowledgments:** The authors express their gratitude to the Ministry of Science, Technological Development and Innovation of the Republic of Serbia, for supporting scientific research, which is essential for the advancement of a knowledge-based society.

**Conflicts of Interest:** The authors declare no conflicts of interest. The funders had no role in the design of the study; in the collection, analyses, or interpretation of data; in the writing of the manuscript; or in the decision to publish the results.

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
