# Peer review of "Development of an Integrated Model for Open-Pit-Mine Discontinuous Haulage System Optimization"

_sustainability, doi:10.3390/su16083156_

Round 1

Reviewer 1 Report

Comments and Suggestions for Authors

Advantages.

Good research in the field of practical mining engineering systems using a fuzzy approach.  A very detailed description of the experimental methodology is given. The topic of this article is interesting and meaningful for the selection of optimal equipment for discontinuous haulage systems.   

The design of the manuscript is well structured:

-          Introduction part with literature analysis is given.

-          The methodology part with algorithms is given.

-          Experimental results and analysis part are given (Case study and Results).

-          Conclusion’s part is given.

There are no significant criticisms about the research methodology. The results part is very convincing and comprehensive.

Disadvantages:

-          Mistake in Line 8 (insert space before your e-mail)

-          Align the equation numbers after (9)

-          Line 215. Is it a mistake – (Figure 7)?

-          Future research directions may also be highlighted in the conclusions part.

-          Table 9. What is L,S,D?

-          It is not clear how the values in Table 7 were obtained. A more in-depth explanation would be needed.

-          Table 7 does not distinguish criteria boundaries, i.e. for example, in line K1 - which numbers refer to criterion K1, which to K2, etc. The same applies to Table 8.

Author Response

Comments and Suggestions for Authors: Rev. 1

Advantages:

Good research in the field of practical mining engineering systems using a fuzzy approach.  A very detailed description of the experimental methodology is given. The topic of this article is interesting and meaningful for the selection of optimal equipment for discontinuous haulage systems.   

The design of the manuscript is well structured:

-          Introduction part with literature analysis is given.

-          The methodology part with algorithms is given.

-          Experimental results and analysis part are given (Case study and Results).

-          Conclusion’s part is given.

There are no significant criticisms about the research methodology. The results part is very convincing and comprehensive.

Disadvantages:

Mistake in Line 8 (insert space before your e-mail)

RESPONSE: Agree. Changes made. Manuscript revised accordingly.

Align the equation numbers after (9)

RESPONSE: Agree. Changes made. Manuscript revised accordingly.

 Line 215. Is it a mistake – (Figure 7)?

RESPONSE: Agree. Changes made. Manuscript revised accordingly.

Future research directions may also be highlighted in the conclusions part.

RESPONSE: Agree. Changes made. Manuscript revised accordingly.

Table 9. What is L,S,D?

      RESPONSE: l, s and d stand for triangular fuzzy number parameters (Figure 2). It is explained in the text and shown in the Fig.

It is not clear how the values in Table 7 were obtained. A more in-depth explanation would be needed.

RESPONSE: In Table 7, a mutual assessment of the criteria was performed, and as written in the text, using a scale of relative importance, or the so-called fuzzified scale where each linguistic variable has its own numerical value. On the other hand, the weighting coefficients were calculated, according to the methodology described in detail, and calculated in a purpose-built application that was mentioned in the text of the paper.

Table 7 does not distinguish criteria boundaries, i.e. for example, in line K1 - which numbers refer to criterion K1, which to K2, etc. The same applies to Table 8.

RESPONSE: According to the journal rule, no lines are placed in the table. The criteria consist of triangular fuzzy numbers, which means that each criterion occupies three columns in the table.

Reviewer 2 Report

Comments and Suggestions for Authors

The authors present an interesting research regarding the selection of mining equipment for discontinuous haulage systems. The subject has been analyzed through numerous scientific publications and technical reports, and the authors aim at enriching the body of knowledge in this field.

The selection of optimal mining equipment for discontinuous haulage systems is one of the most important decisions that need to be made when an open pit mine is designed. There are several influencing factors, including natural, technical, economic, social, etc. They are usually site specific. Some of them are presented numerically while others are rather descriptive and can be stated by linguistic variables. These factors are often related to significant uncertainties, linked to exploration and mining operations. In this context, the experience, knowledge, and expert judgment of engineers and specialists are of key importance for the management of mining processes. The research paper proposes a holistic model that translates into linguistic variables all the criteria that affect the selection of the optimal solution. The authors use the multiple-criteria decision making approach and combining it with fuzzy logic, an algorithm is developed which handle above-mentioned uncertainties. The fuzzy analytic hierarchy process (FAHP) is used in order to deal with the decision problem such as mining equipment and management system selection. The entire algorithm is applied to a real case study.

The overall methodology is well presented providing an adequate theoretical basis. The literature review include relevant references.

The FAHP method has been applied to create the holistic model able of optimizing the selection of optimal equipment for a discontinuous haulage system. The approach has included the definition of relevant criteria for the loading and haulage operations, as well as credible alternatives. The model combines multiple criteria that can be expressed numerically and linguistically offering and integral approach. This integrality is a contribution to the body of knowledge. The approach has been successfully applied for dump truck selection when optimizing the excavator-truck system.

However, the authors should discuss future research works. This part is missing in the presented paper. 

Author Response

Comments and Suggestions for Authors: Rev. 2

The authors present an interesting research regarding the selection of mining equipment for discontinuous haulage systems. The subject has been analyzed through numerous scientific publications and technical reports, and the authors aim at enriching the body of knowledge in this field.

The selection of optimal mining equipment for discontinuous haulage systems is one of the most important decisions that need to be made when an open pit mine is designed. There are several influencing factors, including natural, technical, economic, social, etc. They are usually site specific. Some of them are presented numerically while others are rather descriptive and can be stated by linguistic variables. These factors are often related to significant uncertainties, linked to exploration and mining operations. In this context, the experience, knowledge, and expert judgment of engineers and specialists are of key importance for the management of mining processes. The research paper proposes a holistic model that translates into linguistic variables all the criteria that affect the selection of the optimal solution. The authors use the multiple-criteria decision making approach and combining it with fuzzy logic, an algorithm is developed which handle above-mentioned uncertainties. The fuzzy analytic hierarchy process (FAHP) is used in order to deal with the decision problem such as mining equipment and management system selection. The entire algorithm is applied to a real case study.

The overall methodology is well presented providing an adequate theoretical basis. The literature review include relevant references.

The FAHP method has been applied to create the holistic model able of optimizing the selection of optimal equipment for a discontinuous haulage system. The approach has included the definition of relevant criteria for the loading and haulage operations, as well as credible alternatives. The model combines multiple criteria that can be expressed numerically and linguistically offering and integral approach. This integrality is a contribution to the body of knowledge. The approach has been successfully applied for dump truck selection when optimizing the excavator-truck system.

However, the authors should discuss future research works. This part is missing in the presented paper. 

RESPONSE: Agree. Changes made. Manuscript revised accordingly.

Reviewer 3 Report

Comments and Suggestions for Authors

The application of fuzzy approach to solving problems of qualitative multicriteria analyses has been actively developed in recent years.

The authors have successfully applied fuzzy MCDA  to optimize the selection of mining equipment, specifically the type of dump truck for an existing excavator.  

Four alternative solutions, or four types of dump trucks, were examined and analyzed against several criteria to optimize equipment selection.

But the information provided on criteria and process parameters is so detailed that optimization could be easily performed using traditional mathematical planning methods. Fuzzy modeling is generally used when sufficient information is not available.

As rightly noted by the authors-proposed methodology is universal and can be applied to optimize other parameters of mining operations. 

In this case, the novelty will consist only in the application of the methodology to other processes.

In general, I would like to thank the authors for interesting and fruitful work on the development of fuzzy modeling methods for complex technological processes

Author Response

Comments and Suggestions for Authors: Rev. 3

The application of fuzzy approach to solving problems of qualitative multicriteria analyses has been actively developed in recent years.

The authors have successfully applied fuzzy MCDA  to optimize the selection of mining equipment, specifically the type of dump truck for an existing excavator.  

Four alternative solutions, or four types of dump trucks, were examined and analyzed against several criteria to optimize equipment selection.

But the information provided on criteria and process parameters is so detailed that optimization could be easily performed using traditional mathematical planning methods. Fuzzy modeling is generally used when sufficient information is not available.

As rightly noted by the authors-proposed methodology is universal and can be applied to optimize other parameters of mining operations. 

In this case, the novelty will consist only in the application of the methodology to other processes.

In general, I would like to thank the authors for interesting and fruitful work on the development of fuzzy modeling methods for complex technological processes.

RESPONSE: We would like to express our great gratitude for the positive comments.